# Extracellular Vesicles Are More Potent Than Adipose Mesenchymal Stromal Cells to Exert an Anti-Fibrotic Effect in an In Vitro Model of Systemic Sclerosis

**DOI:** 10.3390/ijms22136837

**Published:** 2021-06-25

**Authors:** Pauline Rozier, Marie Maumus, Claire Bony, Alexandre Thibault Jacques Maria, Florence Sabatier, Christian Jorgensen, Philippe Guilpain, Danièle Noël

**Affiliations:** 1INSERM U1183, Hôpital Saint-Eloi, IRMB, University of Montpellier, 80 Avenue Augustin Fliche, CEDEX 5, 34295 Montpellier, France; p-rozier@chu-montpellier.fr (P.R.); marie.maumus@inserm.fr (M.M.); claire.bony@inserm.fr (C.B.); christian.jorgensen@inserm.fr (C.J.); p-guilpain@chu-montpellier.fr (P.G.); 2Department of Internal Medicine, Multi-Organic Diseases, CHU, 34295 Montpellier, France; a-maria@chu-montpellier.fr; 3INSERM, INRA, C2VN, Aix Marseille University, 13005 Marseille, France; florence.sabatier@ap-hm.fr; 4Clinical Immunology and Osteoarticular Disease Therapeutic Unit, Department of Rheumatology, CHU, 34295 Montpellier, France

**Keywords:** systemic sclerosis, mesenchymal stem cells, TGFβ1, anti-fibrotic, extracellular vesicles

## Abstract

Systemic sclerosis (SSc) is a complex disorder resulting from dysregulated interactions between the three main pathophysiological axes: fibrosis, immune dysfunction, and vasculopathy, with no specific treatment available to date. Adipose tissue-derived mesenchymal stromal cells (ASCs) and their extracellular vesicles (EVs) have proved efficacy in pre-clinical murine models of SSc. However, their precise action mechanism is still not fully understood. Because of the lack of availability of fibroblasts isolated from SSc patients (SSc-Fb), our aim was to determine whether a TGFβ1-induced model of human myofibroblasts (Tβ-Fb) could reproduce the characteristics of SSc-Fb and be used to evaluate the anti-fibrotic function of ASCs and their EVs. We found out that Tβ-Fb displayed the main morphological and molecular features of SSc-Fb, including the enlarged hypertrophic morphology and expression of several markers associated with the myofibroblastic phenotype. Using this model, we showed that ASCs were able to regulate the expression of most myofibroblastic markers on Tβ-Fb and SSc-Fb, but only when pre-stimulated with TGFβ1. Of interest, ASC-derived EVs were more effective than parental cells for improving the myofibroblastic phenotype. In conclusion, we provided evidence that Tβ-Fb are a relevant model to mimic the main characteristics of SSc fibroblasts and investigate the mechanism of action of ASCs. We further reported that ASC-EVs are more effective than parental cells suggesting that the TGFβ1-induced pro-fibrotic environment may alter the function of ASCs.

## 1. Introduction

Systemic sclerosis (SSc) is a severe disease characterized by generalized dysfunctions, including diffuse fibrosis, general vasculopathy, and immune system deregulation [1]. The knowledge of the mechanisms involved in the pathology is even more complex, as SSc is a heterogeneous disease both clinically and biologically. Fibroblasts are widely involved in the physiopathology of SSc, and diffuse fibrosis is the main cause of organ dysfunction. In the pathological environment, fibroblasts actively proliferate, accumulate because of reduced apoptosis, and differentiate into myofibroblasts responsible for exaggerated and uncontrolled production of collagens and extracellular matrix (ECM). Transforming Growth Factor β1 (TGFβ1) plays a major role in fibrogenesis [2]. TGFβ1 is an immunosuppressive and pro-fibrotic cytokine secreted in a latent form, notably by immune cells and sequestered by components such as fibrillin-1 in the ECM [3]. Once activated by the fibroblast-mediated release of integrins or thrombospondin-1, TGFβ1 activates canonical and non-canonical intracellular pathways that induce biological responses, including pro-fibrotic activity [4]. As a consequence, a strong expression of the TGFβ1-responsive gene signature is observed in the skin of patients with severe diffuse cutaneous SSc [5,6]. TGFβ1 stimulation of healthy dermal fibroblasts has been reported to mimic the key characteristics of SSc myofibroblasts [7,8,9,10]. In addition to fibrosis, dysregulation of the immune system plays also a major role [11], particularly in the initial phase of the disease. Macrophages, lymphocytes, or mast cells infiltrate the affected tissues. B lymphocytes seem to be highly activated, as shown by the presence of various autoantibodies in patients. Finally, the endothelium is largely involved in vasculopathy [12]. Endothelial cells enter apoptosis while no compensation by neovascularization is possible, and many mediators of the endothelial function are deregulated, notably endothelin 1, which is produced in excess. 

This complexity certainly explains the obstacles encountered in developing a curative treatment, which is still not available today. Immunosuppressive drugs and hematopoietic stem cell transplantation are two current options to stop the disease course of some selected patients, but these strategies are associated with heavy side effects. To overcome the limitations of current therapeutic options, mesenchymal stromal/stem cells (MSCs) are an attractive alternative approach due to their low immunogenicity and immunosuppressive function. Their additional anti-fibrotic and pro-angiogenic properties make them a promising treatment for SSc patients, thus targeting the three main axes of disease dysfunction. Consequently, several pre-clinical studies have reported their therapeutic effect in murine models of SSc, and clinical trials are in progress [13]. MSCs can reduce fibrosis and improve the inflammation and remodeling-associated molecular signature in the skin and lungs of SSc-induced mice [14]. Interestingly, human MSCs from bone marrow (BM-MSCs) and adipose tissue (ASCs) were shown to be equally effective in a murine model of SSc [15]. MSCs exert their pleiotropic effect by contact with target cells but mainly through soluble mediators released in the extracellular environment or contained within extracellular vesicles (EVs). MSC-derived EVs (MSC-EVs) display the main functions of parental cells and have therefore aroused considerable attention as an alternative therapeutic strategy in many diseases [16]. They are also of interest for SSc treatment, and we have recently demonstrated their beneficial effect in the HOCl-induced murine model of SSc [17].

Even though miR-29a-3p was identified as playing a key role in this model, little is known about the factors that mediate the therapeutic function of EVs in SSc [17]. A better understanding of the mechanism of action involved in the functional properties of MSC-EVs in the context of SSc is therefore particularly relevant for future clinical translation. In the present study, we investigated the anti-fibrotic effect of ASCs, and their derived EVs, using an in vitro model of coculture based on TGFβ1-induced myofibroblasts. This model will be useful to challenge the role of biomolecules with anti-fibrotic activity secreted by MSCs or MSC-EVs using gain- or loss-of-function experiments. 

## 2. Results

### 2.1. TGFβ1 Upregulates the Myofibroblastic Signature in Both Healthy and SSc Human Fibroblasts

Because fibroblasts from SSc patients are rare and difficult to obtain in large quantities, we set up a model of myofibroblasts using TGFβ1-mediated stimulation of dermal fibroblasts from healthy donors (H-Fb). H-Fb were starved for 24 h before TGFβ1 stimulation for an additional 24 h period. Phenotype analysis, as well as proliferation and apoptosis rates, were determined on days 0, 1, and 2 (Figure 1A). TGFβ1-activated H-Fb (Tβ-Fb) displayed a significantly higher proliferation rate on days 1 and 2 compared to H-Fb, while the apoptotic rate tended to be lower (Figure 1B). By contrast to elongated H-Fb, Tβ-Fb appeared flattened and showed a myofibroblast-like cell shape on day 0 and day 1 (Figure 1C). Cytoskeleton modification was also supported by the significant increase in *α-SMA* expression on days 0 and 1 in Tβ-Fb (Figure 1D). Furthermore, Tβ-Fb expressed higher levels of *COL1A1* and lower *MMP3/TIMP2* ratio compared to non-stimulated H-Fb, while the *MMP2/TIMP2 ratio* was significantly increased on day 1. Finally, TGFβ1 stimulation significantly increased the expression of *IL-6*, while *IL-1β* expression tended to be decreased on days 0 and 1. Gene modulation was lower on day 2 (data not shown). Further analyses were therefore performed on day 1.

Similar conditions were used using SSc-Fb isolated from two patients. The morphology of SSc-Fb was already flattened in the absence of TGFβ1 stimulation, but further alterations were observed after stimulation, as showed by the enlarged flattened shape on days 0 and 1 (Figure 2A). 

The expressions of *αSMA*, *COL1A1*, *COL3A1*, *IL-6* were higher in TGFβ1-stimulated SSc-Fb than in SSc-Fb, while the expressions of *MMP1/TIMP1*, *MMP3/TIMP2*, *IL-1β* were significantly lower (Figure 2B). Altogether, the results indicated that TGFβ1 induced a myofibroblastic phenotype in H-Fb and exacerbated the myofibroblastic phenotype of SSc-Fb.

### 2.2. TGFβ1-Induced Healthy and SSc Fibroblasts Display a Similar Myofibroblastic Phenotype

When comparing SSc-Fb and Tβ-Fb, we found no significant difference in the expressions of *αSMA*, *COL1A1*, *MMP1/TIMP1*, *MMP3/TIMP2* on days 0 and day 1 of culture (Figure 3). 

By contrast, TGFβ1 stimulation significantly increased the expression of *αSMA*, *COL1A1* and decreased the *MMP1/TIMP1* ratio in SSc-Fb compared to Tβ-Fb. A similar effect was observed on day 1, even though significance was obtained only for *αSMA* expression. No difference in the expression of the other markers was seen (data not shown). These data pointed out that TGFβ1 stimulation of H-Fb is sufficient to induce a myofibroblastic phenotype close to that of SSc-Fb isolated from patient biopsies suggesting they can be used as a relevant model of myofibroblasts.

### 2.3. ASCs Partly Reversed the Myofibroblastic Phenotype of TGFβ1-Stimulated Fibroblasts

We then investigated the effect of ASCs on the myofibroblastic phenotype of Tβ-Fb in coculture for 24 h. In a first experiment, we evaluated the ratio of ASC:Tβ-Fb that was the most efficient in regulating the expression of myofibroblastic markers in Tβ-Fb. (Figure 4A). As expected, and compared to H-Fb, Tβ-Fb expressed higher levels of *αSMA*, *COL1A1*, *COL3A1*, *MMP2/TIMP2*, *IL-6*, and lower levels of *MMP1/TIMP1*, *MMP3/TIMP2*, *COX2*. The addition of ASCs on Tβ-Fb reversed the expression of several myofibroblastic markers, as shown by the downregulation of *αSMA*, *COL1A1*, and the upregulation of *MMP1/TIMP1*, *COX2* (Figure 4A). No dose-effect was observed since no significant difference was seen, whatever was the ASC:Fb ratio (from 1:1, 1:3, to 1:10), and we, therefore, used the 1:3 ratio in the following experiments.

We then examined the effect of ASCs on both Tβ-Fb- and TGFβ1-stimulated SSc-Fb. ASCs significantly downregulated the expression of *αSMA*, *COL1A1* in both types of stimulated Fb (Figure 4B). In addition, significant upregulation of *MMP1/TIMP1*, *MMP3/TIMP2* was seen in Tβ-Fb, while upregulation of *IL-6*, *COX2* was observed in TGFβ1-stimulated SSc-Fb. Of note, no regulation of gene expression was observed in H-Fb or SSc-Fb that were not stimulated with TGFβ1 (Appendix A). The data indicated that TGFβ1 stimulation was required for fibroblasts, either Tβ-Fb or TGFβ1-stimulated SSc-Fb, to respond to ASC treatment by improving the expression of fibrosis and remodeling markers. 

### 2.4. ASC Secretome and Derived EVs Are More Effective Than Parental Cells for Improving the Myofibroblastic Phenotype

Since we have previously shown that EVs isolated from ASCs can efficiently improve clinical, histopathological, and molecular features of scleroderma in a murine model of SSc, we tested the effect of EVs on Tβ-Fb. First, we used the conditioned supernatant (SN) obtained from ASCs and cultured Tβ-Fb with different doses of SN for 24 h. The SN drastically reversed the myofibroblastic phenotype, as shown by the reduced expression of *αSM-A*, *COL1A1*, *COL3A1*, whereas *MMP1/TIMP1*, *COX2* were increased (Figure 5A). The effect of SN was even higher than that of ASCs, which could not significantly improve some of the markers.

We then used total EVs isolated from the SN of ASCs by differential ultracentrifugation. Different doses of EVs, ranging from 25 to 400 ng of total protein equivalent amounts, were tested. The lowest dose of 25 ng corresponded to the quantity of EVs produced by the number of ASCs added in the coculture, at the ASC:Fb ratio of 1:3, used in Figure 4. All EV doses were able to improve the myofibroblastic phenotype of Tβ-Fb, as shown by the decreased expression of *αSMA*, *COL1A1*, *COL3A1*, *IL-6* and the upregulation of *MMP1/TIMP1* (Figure 5B). Although not significant, upregulation of *MMP2/TIMP2*, *MMP3/TIMP2* was also observed. Of note, some markers were modulated by EVs and not by ASCs, in particular *IL-6*, and the dose of 400 ng EVs might be slightly more efficient than the lower doses. Altogether, these results indicated that ASCs exert an anti-fibrotic effect on myofibroblasts but their derived EVs are more efficient in improving the myofibroblastic phenotype. 

## 3. Discussion

In the present study, we implemented an in vitro model of myofibroblasts to overcome cell shortage from SSc patients. We provided evidence that TGFβ1 enhanced the myofibroblastic gene expression in both H-Fb and SSc-Fb, indicating that a pro-fibrotic environment is required to induce or maintain the myofibroblastic phenotype in vitro. Of note, Tβ-Fb displayed a myofibroblastic signature close to that of non-stimulated SSc-Fb. Furthermore, we showed that EVs isolated from the ASCs exerted anti-fibrotic and pro-remodeling functions on Tβ-Fb, which could, therefore, be useful to investigate the factors contained in EVs that could mediate these functions.

The innovative aspect of our work was to compare Tβ-Fb and SSc-Fb, which are rarely investigated because of the limited access to such samples from patients. We showed that the main markers of fibrosis, namely *α**SMA* and *COL1A1*, as well as the MMP1/TIMP1 ratio, were expressed at similar levels in Tβ-Fb and SSc-Fb, indicating a close myofibroblastic signature and the relevance of the Tβ-Fb model for mimicking SSc-Fb. Moreover, we reported that TGFβ1 stimulation of SSc-Fb further upregulated the expression of fibrotic genes and downregulated that of remodeling and inflammatory markers, suggesting that continuous regulation by TGFβ1 is required to maintain the myofibroblastic profile. TGFβ1 stimulation of healthy dermal fibroblasts to mimic key characteristics of myofibroblasts has already been reported. Different doses of TGFβ1 in the range of 5–20 ng/mL for 24–48 h resulted in similar upregulation of α-SMA at mRNA and protein levels [7,8,9,10]. Even though 48 h of incubation is largely used, some markers were no more deregulated in stimulated fibroblasts, suggesting that 24 h stimulation is more relevant for a larger set of markers [9,18,19,20]. The modulated expression of other ECM components (collagens, MMPs, TIMPs, fibronectin), as well as other factors involved in fibroblast activation such as connective tissue growth factor (CTGF), platelet-derived growth factor receptor (PDGFR), thrombospondin (TSP), or IL-6, was also reported [21,22,23]. Overall, we here reproduced the main characteristics of previous models. Such an in vitro model does not perfectly reproduce the in situ phenotypes of SSc-Fb, since the protein content of TGFβ1-stimulated fibroblasts or SSc-Fb is different [24]. Nevertheless, it allows to partly reflect the disease phenotype and is helpful in investigating the impact of the treatments and clarifying their mechanism of action in the absence of rare fibroblast samples from SSc patients. 

We also demonstrated that ASCs could regulate several markers associated with fibrosis, extracellular matrix remodeling, and inflammation and partly reverse the myofibroblastic phenotype of Tβ-Fb or TGFβ1-stimulated SSc-Fb. Prevention of TGFβ1-induced fibroblast-myofibroblast transition was already reported using murine and human BM-MSCs [25]. Other studies have reported the anti-fibrotic role of ASCs or BM-MSCs on TGFβ1-stimulated fibroblasts from different sources [26,27,28,29,30]. This is also in accordance with our previous results demonstrating that human ASCs were able to stop the progression of the disease and improve both histological and molecular parameters in the skin and lungs of HOCl-induced SSc mice [15]. The model of Tβ-Fb will therefore be useful to identify the factors that mediate the anti-fibrotic function of ASCs such as HGF, which has been proposed to be one main anti-fibrotic factor [26,27,31,32].

Interestingly, we reported here that TGFβ1 stimulation of both H-Fb and SSc-Fb was required to highlight the anti-fibrotic effect of ASCs on fibroblasts. This suggests that TGFβ1-stimulated fibroblasts released soluble mediators that were sensed by ASCs, which, in turn, produced anti-fibrotic factors. This inter-cellular crosstalk was also relevant to the in vivo situation, where ASCs are known to be primed by the pathologic environment and adapt their response via the production of appropriate counteracting mediators (for review, see [33]). Moreover, ASCs modulated most markers in TGFβ1-stimulated SSc-Fb and Tβ-Fb independently on their respective expression levels. This also suggests that TGFβ1-related environmental foot printing of fibroblasts was lost upon the culture and supported the interest of TGFβ1-stimulated H-Fb as a relevant in vitro model to investigate the fibroblast-myofibroblast transition. Our results further demonstrated that this transition is reversible and that ASCs can be key actors in this process. 

The other important finding of our work was the major role that EVs play in the crosstalk between myofibroblasts and ASCs. Indeed, ASC-derived EVs exerted a therapeutic effect on Tβ-Fb by improving the expression of all markers tested, even though not always significantly. This is in accordance with our recently published results demonstrating the in vivo efficacy of ASC-derived EVs in the HOCl-induced model of SSc when injected systemically [17]. Notably, we reported an improvement of fibrotic, remodeling, and inflammatory markers in the skins of the treated mice. Our results were also in agreement with two studies reporting that EVs from the umbilical cord- or BM-derived MSCs can downregulate the expression of α-SMA and type I collagen in TGFβ1-stimulated fibroblasts [34,35]. We showed that EVs are even more effective than ASCs, which has already been reported and proposed to be related to different miRNA content [35]. The lower therapeutic effect of ASCs could also be related to the impact of the pro-fibrotic environment they encountered when co-cultured with Tβ-Fb. The factors secreted by fibroblasts could partly induce a myofibroblastic transition in ASCs, thereby attenuating their therapeutic function.

In conclusion, we here demonstrated that ASC-derived EVs are superior to parental ASCs as regulators of the myofibroblastic phenotype. The in vitro model of TGFβ1-stimulated fibroblasts shares the main markers of SSc-Fb isolated from patients and could be useful to identify factors that mediate the anti-fibrotic function of ASC-EVs. Therefore, our data further supported the interest in using ASC-EVs in an anti-fibrotic therapeutic approach for clinical translation in SSC patients.

## 4. Materials and Methods

### 4.1. Cell Isolation and Expansion

Adipose tissue-derived mesenchymal stromal cells and fibroblasts obtained from healthy subjects, called ASCs and H-Fb, respectively, were isolated from surgical residues obtained after aesthetic liposuction or abdominoplasty. [36]. H-Fb from three donors and SSc-Fb from two donors were used in the study. ASCs were isolated from 9 distinct donors. 

Isolation and characterization of ASCs and Fb have been reported previously [36,37]. ASCs were expanded in α-MEM medium containing 10% fetal calf serum (FCS), 100 μg/mL penicillin/streptomycin, 2 mM glutamine, and 1 ng/mL basic fibroblast growth factor (bFGF; R&D Systems, Noyal Châtillon sur Seiche, France). ASCs were used between passages 1 to 3. Fibroblasts were cultured in DMEM medium containing 10% FCS, 100 μg/mL penicillin/streptomycin, 2 mM glutamine, and sub-cultured when reaching sub-confluency. Fibroblasts were used between passages 3 to 5.

### 4.2. Production and Isolation of EVs

ASCs were seeded at 2 × 10^4^ cells/cm^2^ and cultured for 96 h. The EV-free medium was obtained by recovering the αMEM medium containing 20% FCS after overnight ultracentrifugation at 100,000× *g* at 4 °C. This medium was then diluted to get the production medium containing 3% EV-free FCS. After a wash with phosphate buffer saline (PBS), ASCs were further cultured in the production medium. After 72 h, cells were eliminated from the conditioned supernatant by centrifugation at 300× *g*, 4 °C for 10 min, whereas debris and apoptotic bodies were discarded by centrifugation at 2500× *g*, 4 °C for 25 min. Total EVs were then pelleted by two ultracentrifugation steps at 100,000× *g*, 4 °C for 2 h. EVs were characterized by their size and concentration (by Nanoparticles Tracking Analysis), their structure (by cryo-TEM), and their protein content (by flow cytometry and western blot), as described elsewhere [17]. EVs were used freshly prepared.

### 4.3. In Vitro Model of TGFβ1-Induced Myofibroblasts

H-Fb were seeded at 5 × 10^4^ cells/cm^2^ in DMEM medium containing 10% FCS for 8 h, and then (day-2), they were cultured with DMEM plus 1% FCS for 24 h for cell synchronization. On day-1, an inductive medium consisting of DMEM with 1% FCS and 5 ng/mL TGFβ1 (R&D Systems) was added for an additional 24 h. In the meantime, ASCs were seeded onto polyethylene terephthalate (PET) culture inserts with a pore size of 0.4 µm (Corning, Becton Dickinson, Le Pont-de-Claix, France)) in the proliferative medium for 12 h. The following day (day 0 of treatment), ASCs in culture inserts or different doses of EVs were added to the TGFβ1-induced H-Fb in DMEM medium containing 1% FCS for 24 h (Figure 1A).

### 4.4. Proliferation and Apoptosis Quantification

Fibroblasts were washed once with PBS before evaluating the proliferation and apoptosis rate using the CellTiter-Glo^®^ Luminescent Cell Viability Assay and the Caspase-Glo^®^ 3/7 Assay System, respectively, following manufacturer’s instructions (Promega, Charbonnières-les-Bains, France). White microplates (Cellstar^®^; Greiner Bio-one) were used to measure the luminescent signal on a microplate reader (Varioskan, Thermo Fisher Scientific, Waltham, MA, USA).

### 4.5. RNA Extraction and RT-qPCR

The total RNA was extracted using 350 µL RLT buffer from the RNeasy Mini Kit according to the supplier’s recommendations (Qiagen, Les Ulis, France). The reverse transcription of 300 ng RNA was obtained by M-MLV reverse transcriptase (Thermo Fisher Scientific). Real-time PCR was done on 10 ng cDNA using SYBR Green I Master mix (Roche Diagnostics, Meylan, France) and specific primers (Table 1). Values were normalized to the Ribosomal Protein S9 (*RPS9*) housekeeping gene and expressed as a relative expression or fold change using the respective formulae 2^−ΔCT^ or 2^−ΔΔCT^.

### 4.6. Statistical Analysis

Statistical analyses were performed using GraphPad 8 Prism Software. Data distribution was assessed using the Shapiro–Wilk normality test. A one-sample *t*-test or Wilcoxon test was done when the values displayed a normal distribution or not, respectively. When indicated, the statistical analyses between two groups were compared using the Student’s *t*-test or the Mann–Whitney test when values were parametric or non-parametric, respectively. Data are presented as mean ± SEM.

## Figures and Tables

**Figure 1 ijms-22-06837-f001:**
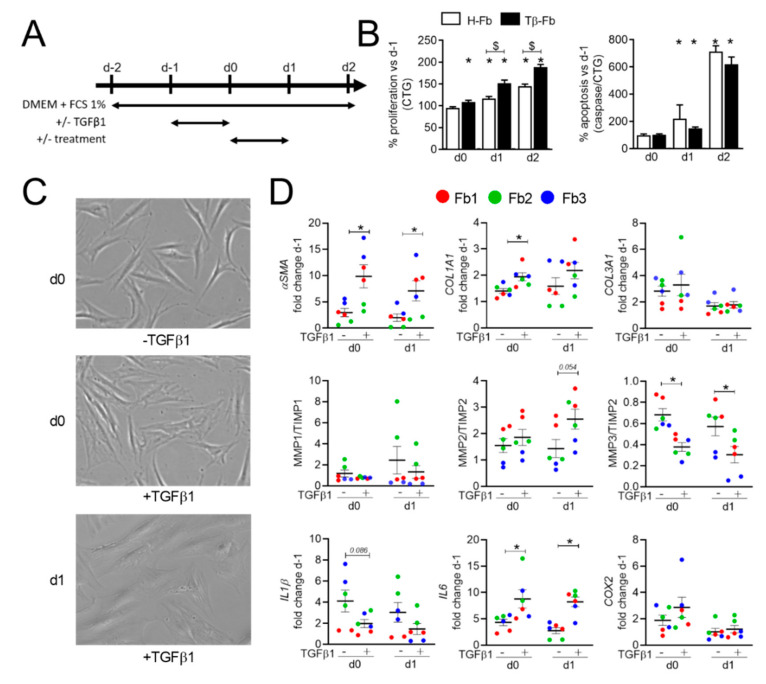
In vitro model of Transforming Growth Factor β1 (TGFβ1)-stimulated fibroblasts. (**A**) Experimental scheme. Dermal fibroblasts from healthy donors (H-Fb) were starved in a DMEM medium containing 1% FCS (day-2) for 24 h before stimulation with TGFβ1 for 24 h. On day 0, treatment was applied for 24 h and samples were analyzed (day 1 or 2). (**B**) The proliferation of H-Fb or TGFβ1-stimulated fibroblasts (Tβ-Fb; left panel) normalized on day-1 (*n* = 12). Apoptosis normalized on CTG assay (right panel). (**C**) Representative pictures of H-Fb (−TGFβ1) on day 0 or Tβ-Fb (+TGFβ1) on days 0 and 1 after stimulation (×40 objective). (**D**) Fold change of gene expression normalized on day-1 (*n =* 6). *: *p <* 0.05 versus H-Fb on day 0 or $: *p <* 0.05 versus the indicated group.

**Figure 2 ijms-22-06837-f002:**
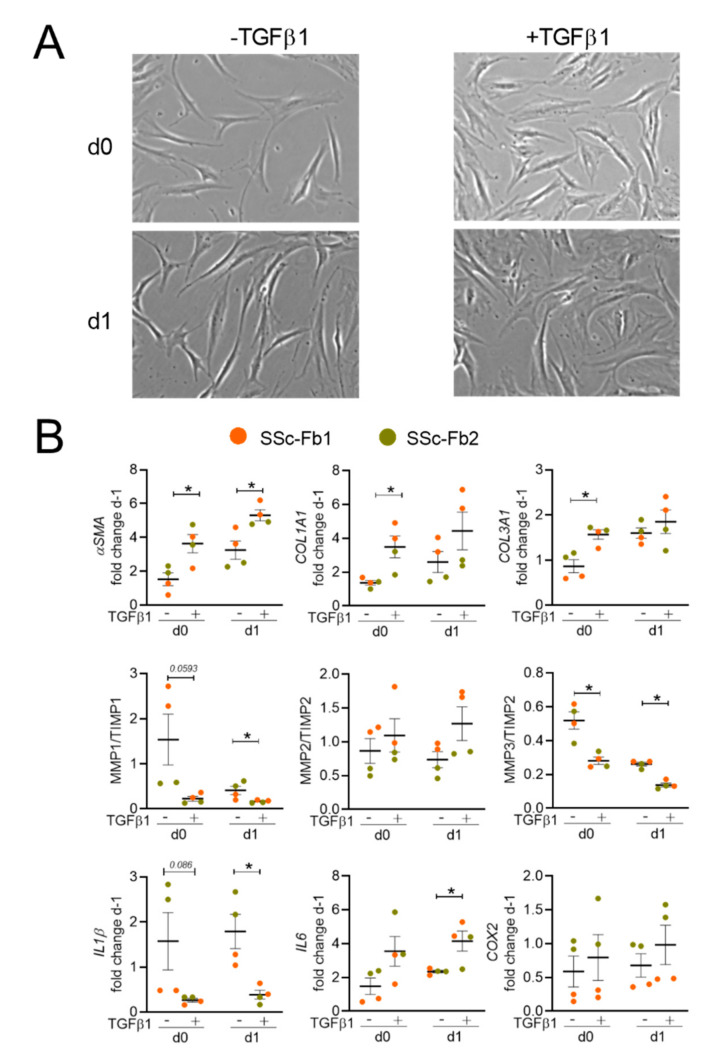
The phenotype of fibroblasts isolated from systemic sclerosis (SSc) patients. (**A**) Representative pictures of fibroblasts from SSc patients (SSc-Fb) cultured with or without TGFβ1 on days 0 and 1 after stimulation (×40 objective). (**B**) Fold change of gene expression normalized on day-1 (*n* = 4). *: *p* < 0.05.

**Figure 3 ijms-22-06837-f003:**
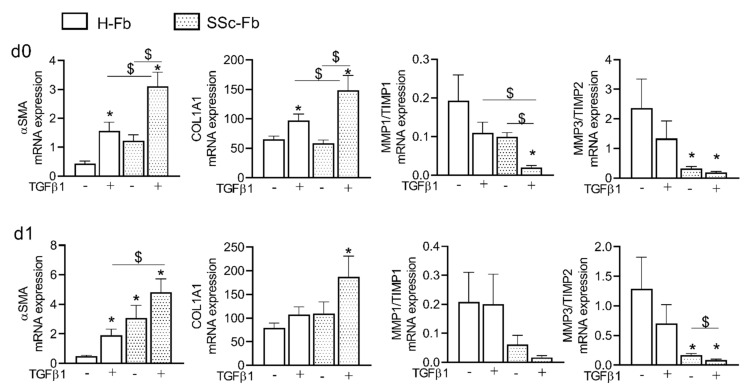
Comparison of the phenotype of healthy and SSc fibroblasts in standard culture conditions or after TGFβ1 stimulation. Fold change of gene expression (*n =* 12). *: *p <* 0.05 compared to H-Fb (−TGFβ1); $: *p <* 0.05 compared to the indicated group.

**Figure 4 ijms-22-06837-f004:**
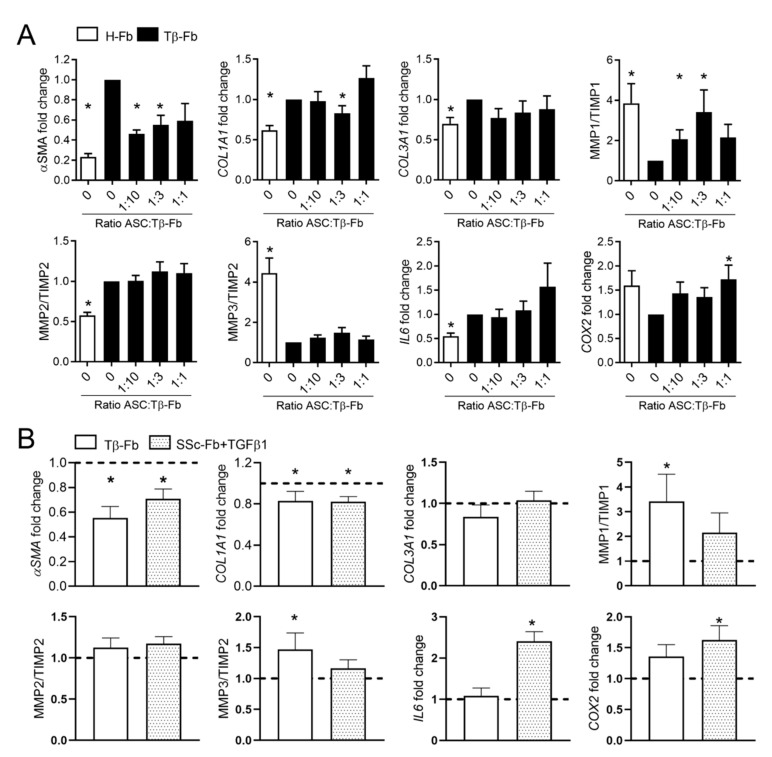
Effect of adipose tissue-derived mesenchymal stromal cells (ASCs) on TGFβ1-stimulated fibroblasts. (**A**) Fold change of gene expression in healthy fibroblasts (H-Fb) and TGFβ1-stimulated fibroblasts (Tβ-Fb) normalized to H-Fb (*n* = 14). (**B**) Fold change of gene expression of markers as in (**A**) in Tβ-Fb or TGFβ1-stimulated SSc-Fb (*n* = 8–14). *: *p* < 0.05.

**Figure 5 ijms-22-06837-f005:**
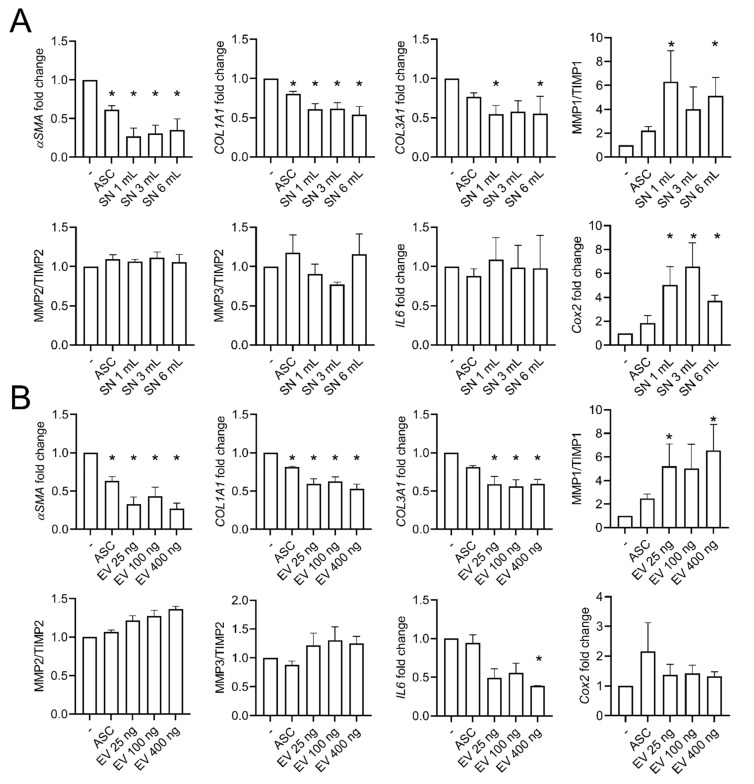
Effects of ASC-EVs on TGFβ1-stimulated fibroblasts. (**A**) Fold change of gene expression in TGFβ1-stimulated fibroblasts using different volumes of conditioned supernatants (SN) from healthy ASCs (*n* = 4). (**B**) Fold change of gene expression as in (**A**) (*n* = 18). *: *p* < 0.05.

**Table 1 ijms-22-06837-t001:** List of primers.

Gene	Forward Sequence	Reverse Sequence
*ACTA2* (αSMA)	CATCGGGATGGAGTCTGCTG	AGAAGCATTTGCGGTGGACA
*COL1A1*	CCTGGATGCCATCAAAGTCT	CGCCATACTCGAACTGGAAT
*COL3A1*	CGCCCTCCTAATGGTCAAGG	AGGGCCTGAAGGACCAGCTT
*COX2*	CGGTGAAACTCTGGCTAGACAG	GCAAACCGTAGATGCTCAGGGA
*IL-1Β*	TGGCTTATTACAGTGGCAATGAGGAT	TCGGAGATTCGTAGCTGGATGCC
*IL-6*	AGACAGCCACTCACCTCTTCAG	TTCTGCCAGTGCCTCTTTGCTG
*MMP1*	AGGCCCAGGTATTGGAGGGGA	GCCGATGGGCTGGACAGGATT
*MMP2*	GCCGTCGCCCATCATCAAGTT	ATAGAAGGTGTTCAGGTATTGCACTG
*MMP3*	GTACCCACGGAACCTGTCCCTC	TTGCGCCAAAAGTGCCTGTCT
*RPS9*	GATTACATCCTGGGCCTGAA	ATGAAGGACGGGATGTTCAC
*TIMP1*	CCGGGGCTTCACCAAGACCT	AGGCAAGGTGACGGGACTGG
*TIMP2*	AGGGCCAAAGCGGTCAGTGA	AACGTCCAGCGAGACCCCAC

## Data Availability

All data are available in the present paper.

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
