# Peer review of "Extracellular Vesicles Are More Potent Than Adipose Mesenchymal Stromal Cells to Exert an Anti-Fibrotic Effect in an In Vitro Model of Systemic Sclerosis"

_ijms, 2021, doi:10.3390/ijms22136837_

Round 1
Reviewer 1 Report
In this amazing work the authors investigated an in vitro model in which Adipose tissue-derived mesenchymal stromal cells (ASCs) and their extracellular vesicles (EVs) are more effective than parental cells for improving the myofibroblastic phenotype.
Could be perfect if the authors might include an anti TGF beta to better verify the experiments
The article is reach of experiments that give a robust conclusion to the article. Although translate this experiments in vivo is not so simple I believe that authors made a real great work
The abstract introduction methods and results are written in a very good way with a logic structure.
The discussion well arguments each points
This article could be accepted in this present form
Author Response
We thank very much the reviewer for the very positive comments on our work. We will think on the possibility to include an anti-TGFB antibody in future experiments.
Reviewer 2 Report
In this paper, TGF-b1-induced model of human myofibroblasts and fibroblasts isolated from SSc patients are employed as samples.
Adipose tissue-derived mesenchymal stromal cells 15 (ASCs) and their extracellular vesicles (EVs) have proved efficacy in this model.
In vitro experiments were conducted in detail.
The specimens used were patient-derived, which is difficult to do in other experimental facilities.
I believe that this is a paper with high originality.
I have a few questions.
In Fig.4, how do you interpret the increase of IL-6 and COX in SSc-fb with ASCs?
IL-6 is thought to be bad in SSc, causing inflammation. Do ASCs cause inflammation?
In Fig.5, what components in EV are causing these suppressive effects? miRNA, cytokines, or other things?
In addition, are EVs used in these experiments within physiological limits?
If excessive, these may cause off-target effects.
To sum up, it is very interesting to note that without TGF, the inhibitory effect of ASCs cannot be obtained.
This may be one of the experimental results that can explain the complex properties of TGFs.
These results are very interesting.
Author Response
In Fig.4, how do you interpret the increase of IL-6 and COX in SSc-fb with ASCs? IL-6 is thought to be bad in SSc, causing inflammation. Do ASCs cause inflammation?
The up-regulation of IL6 and PGE2 has been reported in SSc fibroblasts and related to the development of the pathology. ASCs express IL6 that induces COX2 expression and PGE2 production. PGE2 is an important immunosuppressive factor that has been shown to induce notably the switch of macrophages from M1 to M2 phenotype. PGE2 and IL6 act in a feedback regulatory loop that regulates the tolerogenic response of immune cells but could act differently on other cell types. In our study, it is therefore possible that the secretion of IL6 and PGE2 by ASCs induces the expression of both factors in fibroblasts. Such expression could contribute to an up-regulation of the SSc phenotype, which is not the case since most of the fibrotic and remodeling factors are improved. The resulting overall impact of ASC is therefore beneficial to SSc fibroblasts.
In Fig.5, what components in EV are causing these suppressive effects? miRNA, cytokines, or other things? In addition, are EVs used in these experiments within physiological limits? If excessive, these may cause off-target effects.
Thank you for the interesting questions. We recently reported the role of miR-29a-3p in the beneficial effect of ASC-EV in a in vivo model of SSc (Rozier et al, 2021, J Autoimmunity). The content of EVs has been investigated in another manuscript, which is currently close to be submitted elsewhere. We found out that mRNAs as well as proteins involved in the immunosuppressive function of ASC are also contained within EVs.
As stated in the manuscript, the dose of 25 ng EVs corresponds to the quantity of EVs produced by the number of ASCs added in the coculture, at the ASC:Fb ratio of 1:3. We feel this dose is physiologic even though the physiologic dose is difficult to estimate since EVs produced by the parental cells are continuously recaptured by the parental cells and entrapped by surrounding cells. The fact that increasing doses did not impact the gene expression argues with this.
Identifying the mechanism of action of ASC-derived EVs and the mediators responsible for this beneficial effect is of course very exciting and of upmost importance to understand the role that ASC might play in clinics. This will be the purpose of further research since with this manuscript, we have set up the experimental conditions to study such mechanism. Using loss- or gain-of-function experiments, we should be able to confirm the role played by candidate factors on the phenotype of SSc fibroblasts.
Reviewer 3 Report
In this manuscript Rozier and co-authors show the modulation of Systemic Sclerosis-associated genes in TGFβ1-treated fibroblasts cultured with adipose stem cells (ASC) and the extracellular vesicles secreted by the same cells. The positive effect of both cells and vesicles are clearly presented, but I would say that that novelty of the manuscript is limited, since, as the authors also state, the anti-fibrotic effect of stem cell derived EV has been already reported, and the manuscript follows the work prersented in other papers on the topic.
Unfortunately the positive influence of ASCs wasn't observed in untreated, patients-derived fibroblasts, and this apparently suggests that the model proposed doesn't fully replicates the complexity of the disease.
Nonetheless the authors report here the modulation of several genes in mormal and patient-derived TGFβ1 treated fibroblasts and the paper confirms the hypothesis of anti-fibrotic MSC-derived vesicles.
Of note, TGFβ1-induced conversion of normal fibroblast has been reported years ago, the authors also state this in the discussion of this manuscript. I think that it would be more appropriate to include the information and the references in the introduction.
Author Response
Unfortunately the positive influence of ASCs wasn't observed in untreated, patients-derived fibroblasts, and this apparently suggests that the model proposed doesn't fully replicates the complexity of the disease.
We agree with the reviewer that upon in vitro culture, SSc fibroblasts likely do not maintain their native phenotype because of the absence of the surrounding microenvironment that helps maintaining the pathologic pressure. This is the limitation of in vitro models. The importance of the endothelial compartment has been underlined by many studies. Further studies should consider to include a third party by coculturing endothelial cells with fibroblasts and ASCs. This was however out of the scope of the present study.
Of note, TGFβ1-induced conversion of normal fibroblast has been reported years ago, the authors also state this in the discussion of this manuscript. I think that it would be more appropriate to include the information and the references in the introduction.
Thank you for the suggestion. As recommended by the reviewer, we have included the information and references in the introduction section (please see 15-16, page 3).